# Automated Industrial Composite Fiber Orientation Inspection Using Attention-Based Normalized Deep Hough Network

**DOI:** 10.3390/mi14040879

**Published:** 2023-04-19

**Authors:** Yuanye Xu, Yinlong Zhang, Wei Liang

**Affiliations:** 1Key Laboratory of Networked Control System, Shenyang Institute of Automation, Chinese Academy of Sciences, Shenyang 110016, China; xuyuanye@sia.cn; 2Shenyang Institute of Automation, Chinese Academy of Sciences, Shenyang 110016, China; 3Institutes for Robotics and Intelligent Manufacturing, Chinese Academy of Sciences, Shenyang 110169, China; 4University of Chinese Academy of Sciences, Beijing 100049, China; 5State Key Laboratory of Robotics, Shenyang Institute of Automation, Chinese Academy of Sciences, Shenyang 110016, China

**Keywords:** fiber-reinforced composite, orientation inspection, deep Hough normalization, normalized deep Hough transform, attention-based deep Hough network

## Abstract

Fiber-reinforced composites (FRC) are widely used in various fields due to their excellent mechanical properties. The mechanical properties of FRC are significantly governed by the orientation of fibers in the composite. Automated visual inspection is the most promising method in measuring fiber orientation, which utilizes image processing algorithms to analyze the texture images of FRC. The deep Hough Transform (DHT) is a powerful image processing method for automated visual inspection, as the “line-like” structures of the fiber texture in FRC can be efficiently detected. However, the DHT still suffers from sensitivity to background anomalies and longline segments anomalies, which leads to degraded performance of fiber orientation measurement. To reduce the sensitivity to background anomalies and longline segments anomalies, we introduce the deep Hough normalization. It normalizes the accumulated votes in the deep Hough space by the length of the corresponding line segment, making it easier for DHT to detect short, true “line-like” structures. To reduce the sensitivity to background anomalies, we design an attention-based deep Hough network (DHN) that integrates attention network and Hough network. The network effectively eliminates background anomalies, identifies important fiber regions, and detects their orientations in FRC images. To better investigate the fiber orientation measurement methods of FRC in real-world scenarios with various types of anomalies, three datasets have been established and our proposed method has been evaluated extensively on them. The experimental results and analysis prove that the proposed methods achieve the competitive performance against the state-of-the-art in F-measure, Mean Absolute Error (MAE), Root Mean Squared Error (RMSE).

## 1. Introduction

Fiber-reinforced composites (FRC) have been widely used in various industrial fields, such as aerospace, marine, and sporting goods, thanks to their excellent fatigue and corrosion resistance, super strength, high toughness and light-weight properties [1]. FRC is commonly fabricated by overlapping a certain number of fiber prepreg sheets on the mold with regularly alternating orientations layer-by-layer, as shown in Figure 1. Fiber prepreg sheets are pre-impregnated fibers that have been coated with a resin matrix and then partially cured, making them ready for use in composite fabrication. The fibers are the primary load-carrying element in fiber prepreg sheets, and provide strength and stiffness in the direction of their alignment. Thus, the fiber orientations have a significant impact on the mechanical properties of the final composite parts. However, the automated or manual fiber prepreg sheets placement could bring the discrepancies between the intended fiber orientation and the fiber actual orientation, which may result in the overall system performance degradation. For instance, the deviation of $5^{\circ }$ can reduce the strength of FRC components by almost 20% [2]. Therefore, an efficient fiber orientation inspection system for FRC fabrication is desperately needed if the fiber placement system is not under good control.

Fiber orientation inspection techniques can be broadly classified into two categories: destructive testing and non-destructive testing (NDT) [3]. The destructive testing techniques are typically used in research and development. It focuses on understanding the mechanical properties of the material rather than on manufacturing quality control [4]. NDT includes a range of techniques, such as automated visual inspection [5], ultrasonic testing [6], X-ray radiography [7], and thermography [8,9,10], which allow for the inspection of the fiber orientation without damaging the material. Automated visual inspection uses image processing algorithms to analyze the texture images of FRC and determines the fiber orientation. Compared to other NDT techniques, automated visual inspection can utilize the common 2D camera. It exerts great potentials in measuring fiber orientation, thanks to its advantages in cost-effectiveness, computational efficiency and accuracy.

The Hough transform (HT) and the deep Hough Transform (DHT) are major image processing methods for automated visual inspection of fiber orientation [11,12], as the “line-like” structures of the fiber texture in FRC are readily detectable using these methods [13]. The HT works by mapping image points to a parameter space in which a line can be represented as a single point. In the parameter space, a simple voting scheme is used to identify the most likely line parameters. The DHT employs a deep neural network to perform the mapping from image features to parameter space, which can improve the accuracy, efficiency, and robustness of the HT [11]. However, the DHT still suffers two challenging issues in measuring fiber orientation, which are listed as follows.

(1) Sensitivity to background anomalies: Background anomalies caused by mold regularly appear in FRC images, as shown in Figure 2. The mold background may cause dark spots, lines, or speckles on the FRC images, which could be mixed with fibers and cause the DHT to detect false lines in the image. The reduction of FRC regions in the image also generates more short line segments, further decreasing the line detection performance of DHT.

(2) Sensitivity to longline segments anomalies: The DHT detects a line by searching group of points in an image that lie on the line. Longer line segments containing more points are assigned higher weights in Hough space. However, there are often many longline segments anomalies on the surface of FRC, such as reference line emitted by a laser projector, or gap caused by FRC fracture, as shown in Figure 2. To provide a visible marker for aligning the FRC, the reference line is projected onto its surface. FRC can become damaged or worn over time, which can lead to gaps on the surface. These longline segments anomalies can gain more votes in the Hough space than shorter, true “line-like” structures, which can seriously affect the performance of fiber orientation measurement. Traditional Hough normalization can improve the robustness of the traditional Hough transform to longline segments anomalies [14], but also makes it more sensitive to background anomalies. Because there are many short line segments near the edges of the FRC regions in the images with background anomalies. After traditional Hough normalization, these short line segments may produce peaks in the parameter space that exceed the detection threshold [15], thus reducing the performance of the model in detecting true “line-like” structures.

To address these issues, we propose an attention-based deep Hough network with Hough normalization, which is robust to the background and longline segments anomalies. The main contributions of this paper are summarized as follows.

(1) An attention-based deep Hough network (DHN) is proposed to remove background anomalies caused by mold in FRC images, which seamlessly integrates the attention network and the Hough network together. The attention network is used to learn the important regions of the FRC image that contain the fiber orientation information, while the Hough network is used to detect the fiber orientations in those regions.

(2) The deep Hough normalization is proposed to improve the accuracy and robustness of the DHT when applied to FRC images with longline segments anomalies and background anomalies. It works by normalizing the accumulated votes in the deep Hough space by the length of the corresponding line segment, which completely eliminates the influence of line length on DHT. The deep Hough normalization can make the DHT robust to both longline segments anomalies and background anomalies.

(3) Three separate datasets have been established, which include normal FRC images, FRC images with longline segments anomalies, and FRC images with background anomalies. Our method has been extensively evaluated on these datasets, and achieves the competitive performance against the state-of-the-art in F1, MAE and RMSE.

The rest of this paper is organized as follows: Section 2 summarizes the related works. Section 3 describes the materials and the proposed methods with sufficient detail. Section 4 presents the experimental details and systematic analysis. Section 5 concludes the whole paper.

## 2. Related Work

Automated visual inspection involves using image processing algorithms to analyze the texture images of FRC and determine the fiber orientation. These image processing methods which can be categorized into a few groups namely, statistical methods, spectral methods, structural methods and deep-learning methods. In this section, we briefly review the above methods and their applications in fiber orientation measurement.

### 2.1. Statistical Methods

Statistical methods utilize the analysis of grey-level value distribution surrounding an image pixel to determine patterns. The methods are divided into two categories based on the number of pixels considered to define the local feature: first-order statistics (one pixel) and second-order statistics (two pixels).

Although only the values of individual pixels are considered, first-order methods are often utilized. To measure the orientation of fibers, the structure tensor matrix is commonly employed [16,17]. The structure tensor matrix determines the orientation of a particular region by exploiting the fact that the first-order directional derivatives are greater in directions perpendicular to the fiber orientations than in other directions. The eigenvector associated with the largest eigenvalue of the structure tensor matrix indicates the vertical orientation of the fiber orientation.

In contrast, second-order statistical characteristics depict the connections among the grey-level intensities in a given area. For measuring fiber orientation, numerous second-order techniques have proved to be effective. A grey-level co-occurrence matrix (GLCM) essentially represents the probability distribution of gray levels at positions that satisfy a certain relative distance within an image. Zheng et al. [18] introduced a GLCM-derived parameter that quantifies the extent of diagonal concentration of the GLCM elements and utilized its integration over the relative distance variable as an indicator to identify the dominant texture orientation. Alshehri et al. [19] presented a robust fingerprint descriptor named the co-occurrence of ridge orientations, which encodes the spatial distribution of ridge orientations. The Hessian matrix is created by second-order derivatives. Baranowski et al. [20] suggested a technique based on an eigenvalue analysis of the Hessian matrix in each voxel of a 3D image to determine the local fiber orientation from 3D image data. Pinter et al. [21] compared three methods, including Hessian matrix computation, to evaluate their ability to identify accurate fiber orientations and their susceptibility to errors due to imaging modalities and material composition.

### 2.2. Spectral Methods

Spectral methods can analyze the frequency content of texture strictly in the spatial or frequency domain whose coordinate system has an interpretation closely related to the characteristics of the texture. Spectral methods can analyze the frequency content of texture strictly in the frequency domain (e.g., Fourier transform [22,23]), strictly in the spatial domain (e.g., steerable filters [21,24,25]), or both, frequency and spatial domains (e.g., Gabor [6] and wavelet transform [26]).

The Fourier transform can quantify the spatial variations in an image. The power spectrum of image will have high power in the orientation orthogonal to the fiber orientation. Bardl et al. [22] applied the 2D Fast Fourier Transform (2D-FFT) on the local segments of the eddy current image to determine the local yarn orientation. Brandley et al. [23] proposed a novel method of applying the 2D-FFT to scanning electron microscopy, which map the carbon nanotubes orientation in pre-formed arrays.

The steerable filters have also been used to measure the fiber orientation by locally maximizing the convolutional result of image with a set of orientation-selective filters computed from a linear combination of basis filters. In [21,24], authors compared anisotropic Gaussian filtering method with other methods such as Hessian matrix calculation and structure tensor calculation in their ability to describe correct fiber orientations. They found that the accuracy of the Gaussian filter method depends on the number of sampling directions, which is a clear disadvantage of this method. Legaz-Aparicio et al. [25] proposed a framework based on a simple texture feature extraction step which decomposes the image structures by means of a family of orientated filters.

Gabor and wavelet transform are two other common methods to measure fiber orientation. Yang et al. [6] analyzed the 2D slice of 3D ultrasonic data by the Gabor transform method coupled to the concept of an Information Diagram (GF-ID). The GF-ID improves the Gabor transform method by constructing local Gabor filters with optimal orientation. Lefebvre et al. [26] proposed a method for measuring the texture orientation based on a wavelet analysis, which searching for the orientation that best concentrates the energy of wavelet coefficients in a single orientation.

Statical and spectral methods are rule-based methods and therefore sensitive to noise and anomalies. In addition, they inherently operate in the Cartesian coordinate, which making the accurate extraction of orientation information a secondary process requiring careful correction to remain universally valid [24].

### 2.3. Structural and Deep Learning Methods

Deep-learning methods have also been applied to texture images analysis to measure the fiber orientation, thanks to powerful feature extraction ability. Sabiston et al. [27] used artificial neural networks to predict the fiber orientation within a compression moulded composite by training the network on fiber orientation data. Bleiziffer et al. [28] utilized convolutional neural network (CNN) to compute the direction of the fibers in Micro X-ray computed tomography (CT) images. In addition, they established a workflow based on molecular dynamics simulations to efficiently create synthetic training data for the CNN. Kociołek et al. [29] tested the applicability of CNNs to texture directionality detection and presented the test results focusing on the CNN architectures and their robustness with respect to image perturbations. However, these works directly used the deep-learning model without considering the distribution pattern of texture features.

Structural methods consider texture to be composed of several elements (called primitives), which occur repeatedly according to regular or irregular placement rules. The difference between structural methods is the choice of primitives. Industrial composite fiber surfaces have an obvious “line-like” structure [13]. Therefore, the primitives in this work are line segments. The Hough transform and Radon transform are two classical line detection algorithms. The Hough transform is the discretized form of the Radon transform. Schmitt et al. [30] developed a real-time machine-vision system based on the Hough transform with the main purpose of the real-time, accurate and robust fiber orientation detection under industrial conditions. Holder et al. [31] used the Hough transform to determine the fiber orientation in optical coherence tomography (OCT) images. Nelson et al. [32] used the Radon transform to measure the fiber orientation of local 2D images extracted from the 3D ultrasonic data. The Hough transform and the Radon transform are both sensitive to the image artifacts. To address this problem, the work of [11] proposed deep Hough transform (DHT) by incorporating CNN into the Hough transform. However, the DHT is sensitive to longline segments and background anomalies.

The basic Hough transform also has a bias towards features in the center of the image. The longer lines running through the center of the image have higher potential maxima than lines cutting across the corner of the image. Krieger Lassen [14] introduced normalized Hough transform to overcome this bias. However, the normalized Hough transform not only improves the robustness of the basic Hough transform to longline segments anomalies, but also makes it more sensitive to short lines near the edges of images. Though particular strategies [15] have been introduced to reduce its sensitivity to short lines near the edges of the images, the improvement in performance is limited.

There is currently very limited related work on combining attention networks with the Hough transform. Palme et al. [33] proposed a two-stage Hough transform, which firstly obtained an approximate estimation of lines by applying Hough transform to the entire image, which can be considered as “attention” on the entire image. The second stage then performs a more detailed sampling on specific line segments, which is equivalent to paying “attention” to specific regions. Therefore, although this method does not directly use attention mechanism, its design concept and implementation process can be considered as focusing on specific regions for more detailed processing.

## 3. Materials and Methods

### 3.1. Composite Specimen and Image Acquisition

The fiber prepreg sheet used in this study is CYCOM X850®(Cytec Industries, Woodland Park, NJ, USA). The CYCOM X850 is composed of the thermoplastics toughened epoxy and unidirectional T800 fibers. The resin content of the CYCOM X850 is 35% by weight and the fabric area density is 292 g/m${}^{2}$.

To thoroughly evaluate the robustness of the proposed method in measuring fiber orientation under different real-world scenarios, we establish a normal dataset and two anomalous datasets, each for one type of anomalies (longline segments anomalies, background anomalies). Each dataset has 5400 training images, 3600 validation images, and 5400 test images.

The normal dataset contains 14,400 normal FRC images with 180 different fiber orientations ranging from 0° to 179° at 1° increments, and 80 types of illuminations which cover a large range of light conditions. In order to collect the normal dataset, a data collection platform has been built. The FRC data collection platform consists of four parts: camera, symmetric light source, computer, and two pieces of fiber prepreg sheets. The camera is a 2D camera (HikVision MV-CA050-12GC2) with lens (HikVision MVL-KF2524M-25MP, focal length: 25 mm), connected to the laptop by USB cable. The laptop is equipped with Intel Core i7 processing unit and 16G memory. Different light intensities can be obtained by adjusting the light source height, illumination angle, and light intensities. The ranges of light source heights relative to the FRC layer are [10 cm, 20 cm, 30 cm, 40 cm, 48.5 cm], and the ranges of the illumination angles are [20°, 45°, 75°, 90°]. The acquired images are RGB images with resolution of 2048 × 2048. After being converted to grayscale images, the acquired images are resized to 200 × 200 to compose the normal dataset.

It is well-known that training neural networks can be a tedious task, due to the high cost of obtaining the large amount of labeled data required to train a general model [28]. Utilizing synthetic data for model training has become more popular in recent years as a way to address the issue, which can be obtained quickly and at much lower cost [34]. Both the longline dataset and the background dataset contain a large number of synthetic images generated from the normal dataset.

The longline dataset contains FRC images with longline segments anomalies, such as reference lines emitted by a laser projector or gaps resulting from FRC fractures. The reference lines serve as a precise alignment of the fiber orientation in the actual lay-up scene, and FRC typically fractures along the direction of the fibers. Thus, the orientations of the reference lines or gaps should not deviate too much from the fiber orientation. On the longline dataset, the orientation of the reference lines and gaps deviates no more than ±10° from the fiber orientation. The process of simulating a gap resulting from FRC fractures involves drawing two parallel straight lines on the normal FRC image, filling the space between the lines with white to create a gap, and normalizing the gap to match the brightness of the surrounding FRC material. The gap widths are between 2 to 10 pixels.

The background dataset comprises FRC images with background anomalies caused by mold, which may create dark spots, lines, or speckles on the FRC images. The mold used in this study is a matte metal mold. To create a FRC image with background anomalies, the first step is to generate a binary mask, which can be divided into two parts. Then, we fill one part of the mask with the normal FRC image and the other part with the mold image. Finally, the mold image part is normalized to match the brightness of the FRC image, which ensures that the final synthetic image has a uniform level of illumination across both the FRC and mold regions. Two algorithms are employed to generate the binary mask, which include 1D random walks and B-splines. The area of the normal FRC part accounts for 20% to 40% of the total image area.

### 3.2. Preliminaries

The texture of FRC image consists of long parallel lines. The fiber orientation measurement for the FRC image is to detect the orientation of these parallel lines. As shown in Figure 3, given a 2D image $I_{H\times W}\in R^{H\times W}$, where *H* and *W* are the spatial size, we set the center of the image as the origin. A straight line *l* is one representative fiber texture, which satisfies the following equation in the polar coordinate frame:(1)$$r_{l}=x\cos\theta_{l}+y\sin\theta_{l}$$
where $r_{l}\in \left[-\sqrt{W^{2}+H^{2}}/2,\sqrt{W^{2}+H^{2}}/2\right]$ is the distance from the origin to the line *l*, $\theta_{l}\in \left[0,\pi \right)$ is the angle between the normal of the line *l* and x-axis. We define $\theta_{l}$ as the fiber orientation of the FRC image.

As described in Equation (Equation 1), every straight line in the image can be represented by two parameters θ and *r*. The two parameters are quantized into discrete bins. The discrete binned parameter space is denoted as the Hough space. We define the quantization interval for *r* and θ as Δr and Δθ, respectively. Then the Hough space size, denoted with Θ and *R* can be formulated below:(2)$$\Theta =\frac{\pi }{\Delta \theta },R=\frac{\sqrt{W^{2}+H^{2}}}{\Delta r}$$

### 3.3. Attention-Based Normalized Deep Hough Network

The attention-based normalized Deep Hough Network (AttNorm-DHN) mainly contains four components: (1) an encoder network that extracts pixel-wise deep representations; (2) an attention network that learns the important regions of the FRC image which contains the fiber orientation information; (3) the Norm-DHT that converts the deep representations from the spatial space to the Hough space; (4) the orientation measurement module that measures the fiber orientation in the Hough space. The AttNorm-DHN is an end-to-end framework, which allows the network to learn the entire process directly from raw inputs to the desired outputs. During the training process, we optimize all the components from scratch without any pre-training or transfer learning. The pipeline of AttNorm-DHN is shown in Figure 4.

#### 3.3.1. Encoder Network

The encoder network is an important component that is responsible for extracting pixel-wise deep representations from the input image. The deep representations obtained from the encoder network have a higher level of abstraction compared to the raw pixel values of the input image. These representations are then used to capture the structural information of the image, which is crucial for accurately measuring the fiber orientation in FRC images.

Given an input image $I_{H\times W}\in R^{H\times W}$, the deep CNN features $X\in R^{C\times \frac{H}{2}\times \frac{W}{2}}$ is firstly extracted with the encoder network, where *C* indicates the number of channels and *H*, *W* are the spatial size.

To perform a fine-grained deep Hough transform on the deep CNN features *X*, it is necessary to ensure that the feature resolution of *X* is sufficiently high. To achieve this, a shallow encoder network is designed. Table 1 shows the network architecture of encoder network. The network mainly consists of multiple 3×3 convolution layers except for the last Conv 1×1 layer, which compresses the output from 128 dimensions to 64 dimensions in order to obtain more compact feature representations. We use ReLU activation in all layers of the encoder network [35]. The Maxpool operation correspond to 2×2 maxpooling, which reduces the spatial resolution of the feature maps by a factor of 2 [36].

#### 3.3.2. Attention Network

The attention network takes the FRC image as input and produces an attention map. The attention map is a weight map indicating the importance of each spatial location in the output features of encoder network. It highlights the regions of the image that are most relevant for predicting the fiber orientation.

Given an input image *I*, the attention weights $A\in R^{C\times \frac{H}{2}\times \frac{W}{2}}$ of input image is generated with the attention network. Then the attention weights *A* is used to compute the weighted deep CNN features $X_{att}$.
(3)$$X_{att}=A⊗X$$
where ⊗ indicates the element-wise multiplication.

Table 2 shows the network architecture of attention network. The network architecture starts with two 3×3 convolutional layers with ReLU activation functions. A maxpooling layer is then used to reduce the spatial resolution of the feature maps by a factor of 2, producing 64 feature maps of size 100×100.

After that, a bottleneck-like network [37] is added on top of it to obtain the attention weights. Specifically, the bottleneck-like network consists of four sets of convolutional layers, each comprising two 3×3 convolutional layers with ReLU activation functions and a maxpooling or upsampling layer. The upsampling operations correspond to 2×2 element replication, which increases the spatial resolution of the feature maps by a factor of 2. The first two sets of convolutional layers utilize maxpooling layers to gradually reduce the spatial resolution and increase the number of feature maps. The last two sets of convolutional layers employ upsampling layers to gradually increase the spatial resolution and reduce the number of feature maps.

Finally, the attention network utilizes two 3×3 convolutional layers with ReLU activation functions and a 1×1 convolutional layer with a Sigmoid activation function to obtain the attention weights.

#### 3.3.3. Feature Transformation with Normalized Deep Hough Transform

The deep Hough transform (DHT) can transform lines in the image space into more compact representations in the parameter space [11], which makes the lines easier to detect by neural network. The normalized deep Hough Transform (Norm-DHT) is an improved DHT, which takes advantages of the deep Hough Normalization to further improve the accuracy and robustness of the DHT in the presence of longline segments anomalies and background anomalies.

The Norm-DHT takes $X\in R^{C\times \frac{H}{2}\times \frac{W}{2}}$ as an input and produces the normalized deep Hough features $Y_{norm}\in R^{C\times \Theta \times R}$, where Θ and *R* are the parameters in the Hough space, as described in Equation (Equation 2). Different from the traditional Hough transform, DHT is used to vote in the deep spatial space. Each bin in the deep Hough space corresponds to the features along a straight line in the deep CNN features *X*. Features of all pixels along line *l* in *X* are aggregated to the corresponding position $\left(\theta_{l},r_{l}\right)$ in *Y*:(4)$$Y\left(\theta_{l},r_{l}\right)=\sum_{i\in l}X(i)$$
where *i* is the position index. $\theta_{l}$ and $r_{l}$ are the parameters of line *l*, according to Equation (Equation 1).

Similar to traditional Hough normalization [14], the deep Hough normalization is typically implemented as an additional step following the DHT. After the deep Hough features *Y* is generated, the length of each line segment in the deep CNN features *X* is calculated. Each bin in the deep Hough features *Y* are then divided by the length of the corresponding line segment to perform Hough normalization, as shown in Figure 5.

The length matrix $L\in R^{\Theta \times R}$ can be computed by applying the DHT to an all-ones matrix $J\in R^{H\times W}$ with every pixel set at unity.
(5)$$L\left(\theta_{l},r_{l}\right)=\sum_{i\in l}J(i)$$
where $L\left(\theta_{l},r_{l}\right)$ is the length of the line *l*. $\theta_{l}$ and $r_{l}$ are the parameters of line *l*.

Afterwards, the normalized deep Hough features $Y_{norm}\in R^{C\times \Theta \times R}$ is obtained by dividing deep Hough features *Y* with the length matrix *L*.
(6)$$Y_{norm}\left(\theta_{l},r_{l}\right)=\frac{Y(\theta_{l},r_{l})}{L(\theta_{l},r_{l})}$$

At last, two 3 × 3 convolutional layers with ReLU activation functions and a 1 × 1 convolutional layer with a Sigmoid activation function are applied to yield the value in the Hough space $Z\in R^{\Theta \times R}$. The network architecture of Norm-DHT is shown in Table 3.

#### 3.3.4. Orientation Measurement Module

In this work, we need to measure the orientation θ in the final Hough space regardless of the parameter *r*. A simple method is employed to get the orientation vector from the final Hough space. The orientation vector $Or\in R^{\Theta }$ is obtained by accumulating the final Hough space $Z\in R^{\Theta \times R}$ along the direction of *R*:(7)$$Or=\sum_{i=0}^{R}Z_{\Theta }^{i}$$
where *i* is the positional index along the direction of *R*.

#### 3.3.5. Loss Function

Note that we discretize the Hough space and obtain the orientation vector by accumulating Hough space along the direction of *R*. Thus, the orientation measurement could be converted to the classification, i.e., to predict a probability for each orientation bin. This classification problem can be addressed with the standard cross-entropy loss to simply predict each orientation class independently:(8)$$L=-\frac{1}{m}\sum_{i=1}^{m}y_{i}\log\left(Or_{i}\right)+\left(1-y_{i}\right)\log\left(1-Or_{i}\right)$$
where *m* is the number of training samples. ${Or}_{i}$ is the predicted orientation vector of sample *i*, and $y_{i}$ is the ground-truth orientation vector of sample *i*. We train the end-to-end AttNorm-DHN model by minimizing the loss *L* via the standard Adam optimizer [38].

## 4. Experiments

### 4.1. Experimental Setups

#### 4.1.1. Evaluation Metric

The FRC orientation is a continuous variable and can be quantized to the discrete bins. Thus, the FRC orientation measurement can be viewed as both a regression task and a multi-class classification task. In this work, both the classification and regression evaluation criteria are utilized to assess the performance of the model, including the F−measure, Mean Absolute Error (MAE), Root Mean Squared Error (RMSE).

The macro−F1 score is utilized as our F−measure, which is more robust towards the error type distribution. For a specific orientation class *C*, true positive (TP) indicates the number of images of class C that the model is correctly predicted; false positive (FP) indicates the number of images of class *C* that the model is incorrectly predicted; true negative (TN) corresponds to the number of images that are correctly predicted as not belonging to class *C*; false negative (FN) corresponds to the number of images that are misclassified as class *C*. The equations for macro−F1, MAE, RMSE are presented in Equations (Equation 10)–(Equation 12), respectively.
(9)$$P=\frac{TP}{TP+FP},R=\frac{TP}{TP+FN}$$
(10)$$F1=\frac{2PR}{P+R},macro-F1=\frac{1}{n}\sum_{i=0}^{n}F1_{i}$$
(11)$$MAE=\frac{1}{m}\sum_{j=1}^{m}\arccos(|\cos\left(Or_{j}-y_{j}\right)|)$$
(12)$$RMSE=\sqrt{\frac{1}{m}\sum_{j=1}^{m}{(\arccos(|\cos\left(Or_{j}-y_{j}\right)\left|)\right)}^{2}}$$
where ${F1}_{i}$ is the F1−score of class *i*, ${Or}_{j}$ is the predicted class of sample *j*, and $y_{j}$ is the ground-truth class of sample *j*.

#### 4.1.2. Implementation Details

All images are wrapped into a mini-batch of 64, and the batch normalization is adopted in all the networks as well. We train all models using Adam optimizer for 500 batches in longline dataset and normal dataset, and for 1000 batches in background dataset [38]. The learning rate and momentum are set to $1\times 10^{-3}$ and 0.9, respectively. The quantization interval Δθ is set to 1.

We use a single Nvidia Tesla V100 32GB GPU for training, which is a high-end GPU commonly used for deep learning applications. During the inference phase, a lower-end K80 GPU is utilized to provide a more realistic estimate of the model’s performance when more powerful GPUs may not be available. The model requires 1630 MB of GPU memory during inference, which is within the capacity of most modern GPUs. The inference time on a K80 GPU is 13ms, which is also relatively fast. Overall, it appears that the proposed method has relatively low computational requirements, making it potentially feasible to train and inference in a wide range of applications.

### 4.2. Quantization Interval Tunings

The quantization intervals Δθ and Δr in Equation (Equation 2) play an important role in both the classification performance and the computational efficiency of the AttNorm-DHN. Smaller quantization intervals lead to more quantization levels, which in turn results in larger computational overhead. In this work, we set the angular quantization interval Δθ=1 and train the AttNorm-DHN with different distance quantization interval Δr on the normal dataset. As shown in Figure 6, MAE and RMSE Initially decrease rapidly and then reach a relatively stable level. The turning point is near $\Delta r=\sqrt{2}$. The F-measure firstly rises rapidly, and then stabilizes at a relatively constant level. The turning point is also near $\Delta r=\sqrt{2}$. Thus, the turning point $\Delta r=\sqrt{2}$ is a proper choice for distance quantization, while considering the performance and computational overhead.

### 4.3. Comparison with the State-of-the-Art Methods

We compare the AttNorm-DHN with three methods, including the classical Hough transform method with Canny edge detector (Canny + HT) [31], DHN, and Norm-DHN. The Hough transform method with Canny edge detector involves three main steps: (1) Performing edge detection using the Canny edge detector. (2) Applying the classical Hough transform to detect lines. (3) Calculating the median of the angles of all detected lines. The Norm-DHN is obtained by removing the attention branch from the AttNorm-DHN. The DHN is built by replacing the Norm-DHT component in Norm-DHN network with DHT.

#### 4.3.1. Comparison of the Evaluation Metrics

Table 4 shows the performance of different methods for measuring fiber orientation in FRC on three separate datasets: normal dataset, longline dataset, and background dataset.

On the normal dataset, the DHN method consistently outperforms Canny+HT with a considerable margin. The Norm-DHN and AttNorm-DHN methods achieve slightly higher F-measure scores compared to DHN. However, there is a slight increase in both MAE and RMSE scores for Norm-DHN and AttNorm-DHN compared to DHN. Overall, the DHN, Norm-DHN, and AttNorm-DHN models all demonstrate impressive performance on the normal datasets.

On the longline dataset, the Canny+HT method fails to function, achieving only 0.0158 in F-measure, as it only detects the longline segments anomalies. On the other hand, the DHN approach achieves a substantially higher F-measure score of 0.970, regardless of the relatively higher MAE and RMSE. The Norm-DHN enhances the performance of the DHN, which indicates the robustness of deep Hough normalization to longline segments anomalies. The AttNorm-DHN slightly reduces the performance of Norm-DHN, but is still outperforms the DHN.

On the background dataset, the DHN outperforms the Canny+HT approach by a large margin, which consistently demonstrates the superiority of deep-learning based methods on anomalous datasets. The Norm-DHN tend to perform better than the DHN, because it removes some short lines near the edges of the FRC regions in the images with background anomalies. The AttNorm-DHN achieves the best performance among all other methods.

To summarize, the DHN is sufficient for normal FRC images, the Norm-DHN is adequate for longline segments anomalies, and only the background anomalies necessitate the use of AttNorm-DHN. However, it is worth noting that AttNorm-DHN also performs well on the normal and longline datasets.

#### 4.3.2. Comparison of the Final Hough Space

As defined in Equation (Equation 7), the predicted orientation vector is obtained by accumulating the final two-dimensional Hough space. For the purpose of showing which lines our models focus to measure the orientation, the final Hough space heatmaps of DHN, Norm-DHN, and AttNorm-DHN are visualized in Figure 7. A straight line Θ=θ in the Hough space represents a series of parallel lines of angle θ in the image, where θ is the orientation of the FRC image.

On the normal dataset, the Hough spaces of all methods show almost the same patterns, where there are distinct straight lines with an equation of Θ=θ. This indicates that all the models perform well on the normal datasets. On the longline dataset, the Hough space of Norm-DHN and AttNorm-DHN have less noise than the Hough space of DHN. Additionally, the straight line Θ=θ in the Hough space of Norm-DHN is more noticeable compared to the Hough space of AttNorm-DHN. These results suggest that the Norm-DHN exhibits superior performance relative to the other methods. On the background dataset, the Hough spaces of DHN have global noise that permeates across the entire space, which means that the DHN mistakenly detect numerous straight lines whose angle is not θ. Through the sequential incorporation of the Norm-DHT and attention network to the DHN, the global noise in the Hough space decreases gradually until it nearly disappears.

### 4.4. Importance of Deep Hough Normalization for Attention

Both deep Hough normalization and attention network can improve the performance of the DHT on images with background anomalies. We investigate whether incorporating deep Hough normalization into the attention network can further enhance its performance on the background dataset. To construct the attention-based Att-DHN network, the DHT module is substituted with the Norm-DHT module in the AttNorm-DHN. The evaluation metrics of DHN, Att-DHN, and AttNorm-DHN on the background dataset are summarized in the Table 5. As expected, Att-DHN outperforms the DHN by a large margin, and AttNorm-DHN exhibits modest improvements over the Att-DHN. It demonstrates that incorporating the deep Hough normalization improves the performance of the attention-based Att-DHN as well.

To explore why integrating deep Hough normalization enhances the performance of attention network, the attention heatmaps of Att-DHN and AttNorm-DHN on the background dataset are visualized in Figure 8. The Att-DHN gives more attention to the edges of the FRC regions in the images with background anomalies, resulting in insufficient focus on the important regions that contain fiber orientation information. On the other hand, by incorporating deep Hough normalization into the attention architecture, the AttNorm-DHN exhibits a superior ability to precisely identify and localize these critical regions.

### 4.5. Weakly Supervised Semantic Segmentation

Weakly supervised semantic segmentation is a challenging task that only takes image-level labels as supervision but produces pixel-level predictions for testing [39]. In this paper, we take orientation labels as the supervision to distinguish the mold and FRC regions. As mentioned in the previous section, the AttNorm-DHN can precisely identify and localize FRC regions. We set the binary threshold 0.5 and obtain the binarized masks from the heatmaps of AttNorm-DHN. As shown in Figure 9, the masks predicted by the AttNorm-DHN exhibit discontinuity at the boundary between two regions, appearing less smoothness compared to the ground-truth masks.

In order to quantitatively evaluate the semantic segmentation performance of the AttNorm-DHN on background dataset, we choose the Intersection over Union (IOU) and Pixel Accuracy (PA) as the evaluation metrics [40]. We choose the classical U-net Network for comparison with AttNorm-DHN [37].

In Table 6, it can be seen that U-net is a highly effective fully-supervised semantic segmentation method. However, obtaining pixel-level labels required by the fully-supervised model can be difficult in real-world scenarios, and only image-level labels can be obtained. On the other hand, AttNorm-DHN is a weakly-supervised semantic segmentation method that only requires image-level labels. It can achieve comparable performance to the supervised U-net model, as shown in Table 6. Therefore, considering cost and other practical issues, AttNorm-DHN can serve as a good replacement model for U-net in real-world scenarios.

## 5. Conclusions and Future Works

In this paper, the deep Hough normalization has been proposed for fiber orientation inspection. It normalizes the accumulated votes in the Hough space by the length of the corresponding line segment, making it easier for DHN to detect short, true “line-like” structures. In addition, we have designed an attention-based DHN that integrates attention and Hough networks to remove background anomalies caused by mold in FRC images. Finally, three datasets have been established to evaluate the proposed methods, and our proposed methods have been evaluated extensively on them. The experimental results and analysis revealed that the proposed method achieves the competitive performance against previous arts in terms of F-measure, MAE and RMSE in real-world scenarios with various types of anomalies.

There are several FRCs need further investigation to fully explore the applicability and robustness of the proposed method in different scenarios, including unidirectional fibers with waviness or crimp, transparent (glass) fibers, and woven fabrics with voids and cross-overs in them. These studies aim to enhance the method’s robustness and applicability across various materials, contributing to the development of high-performance composites and advanced materials.

## Figures and Tables

**Figure 1 micromachines-14-00879-f001:**
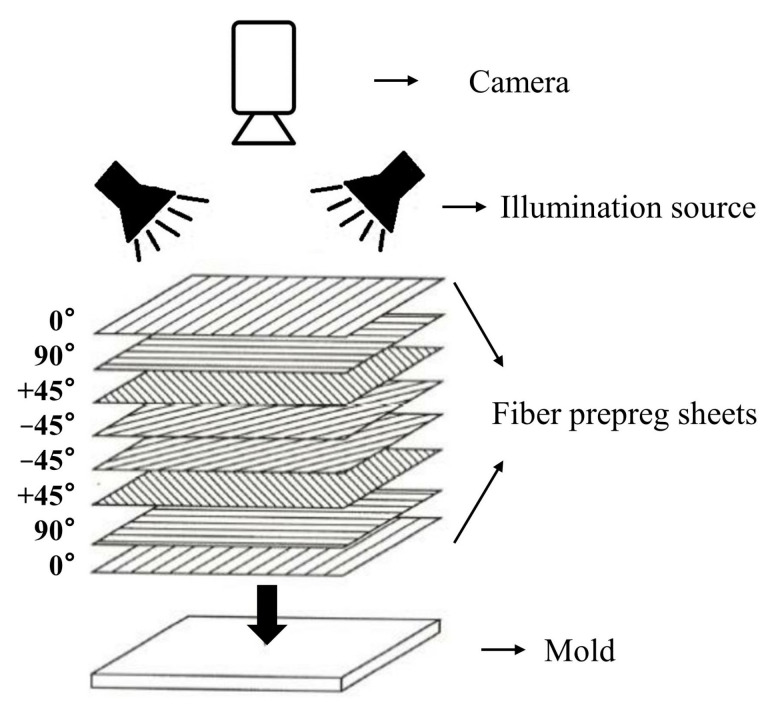
Automated visual inspection system of industrial composite fiber orientation.

**Figure 2 micromachines-14-00879-f002:**
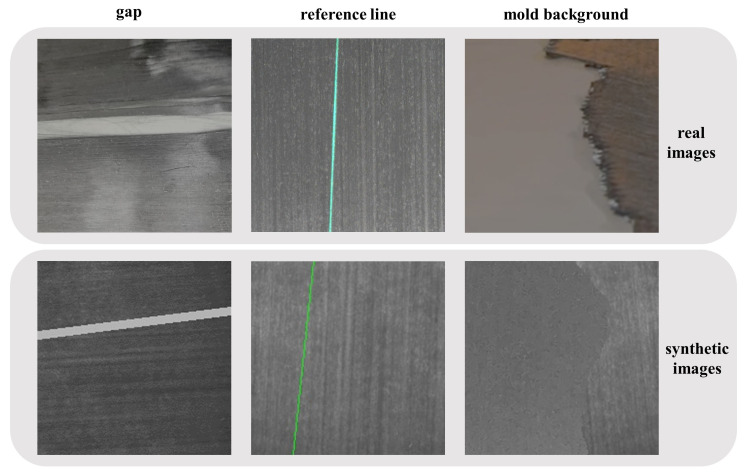
Typical FRC images with various types of anomalies, including the longline segments anomalies caused by gap and reference line, and the background anomalies resulting from mold. In the set of images presented, the first row depicts real FRC images with various type of anomalies, and the second row illustrates corresponding synthetic images. The synthetic images are generated to augment the real images, considering the difficulty in collecting a large number of anomalous images from real-world scenarios. It can be observed that the synthetic images closely resemble the real images.

**Figure 3 micromachines-14-00879-f003:**
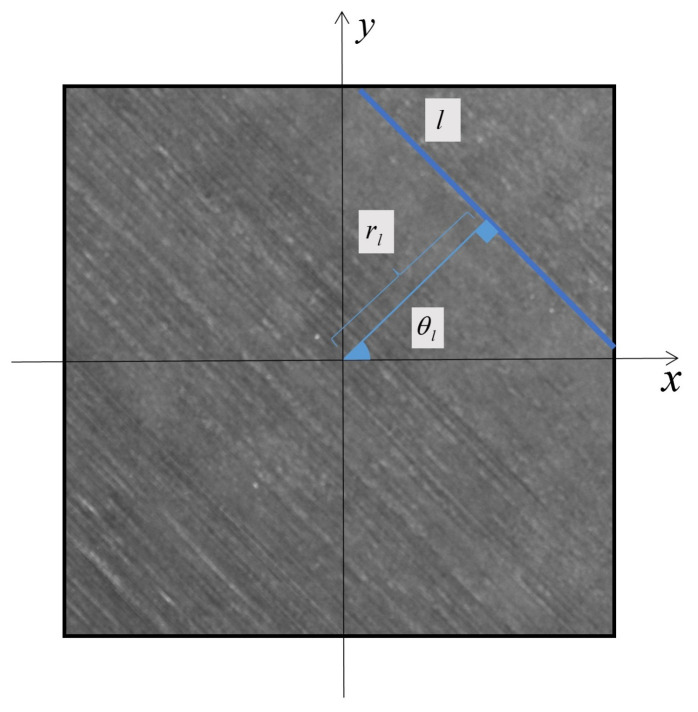
A straight line *l* is one representative long line among the texture of FRC image. $\theta_{l}$ is the angle between the normal of the line *l* and x-axis, $r_{l}$ is the distance from the origin to the line *l*. We define $\theta_{l}$ as the fiber orientation of the FRC image.

**Figure 4 micromachines-14-00879-f004:**
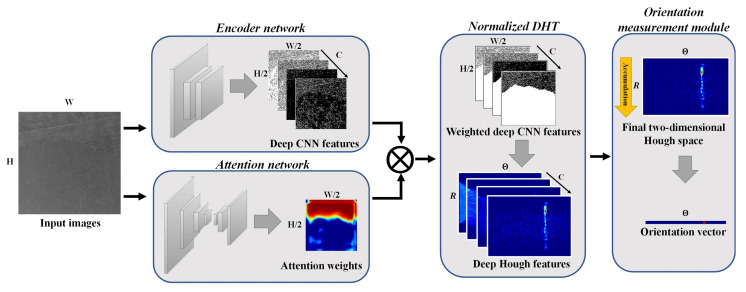
The pipeline of the attention-based normalized Deep Hough Network (AttNorm-DHN). The network consists of four components, i.e., the encoder network, the attention network, the normalized DHT, and the orientation measurement module.

**Figure 5 micromachines-14-00879-f005:**
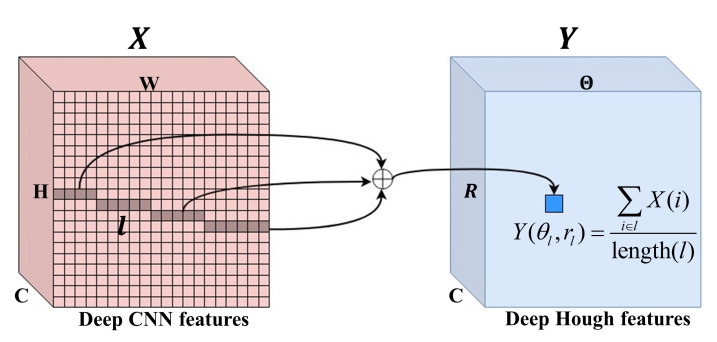
Illustration of the feature transformation with Norm-DHT.

**Figure 6 micromachines-14-00879-f006:**
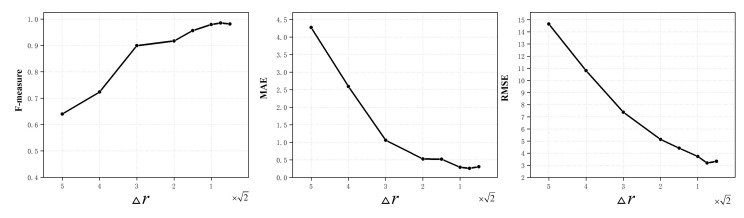
Performance of AttNorm-DHN under different distance quantization intervals Δr.

**Figure 7 micromachines-14-00879-f007:**
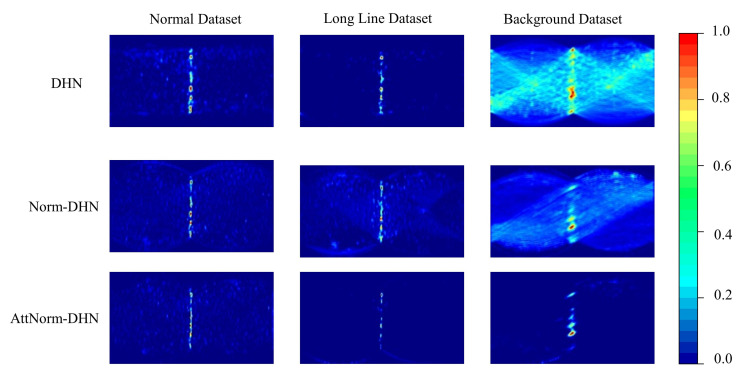
Illustrations on the comparisons of the final Hough space among DHN, Norm-DHN and AttNorm-DHN on three separate datasets.

**Figure 8 micromachines-14-00879-f008:**
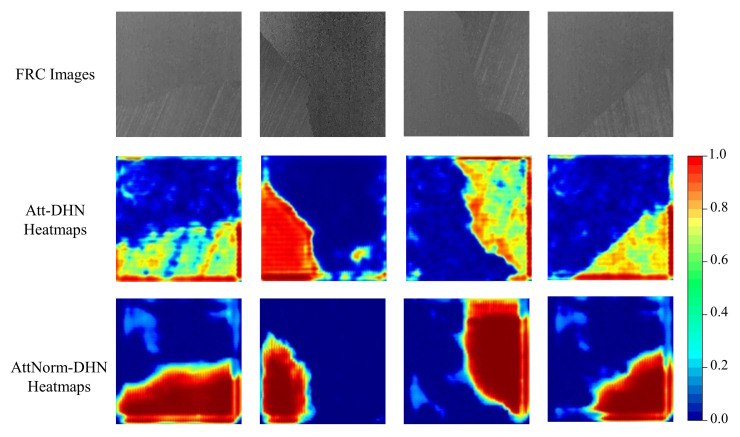
The attention heatmaps of both Att-DHN and AttNorm-DHN alongside their corresponding original images on the background dataset.

**Figure 9 micromachines-14-00879-f009:**
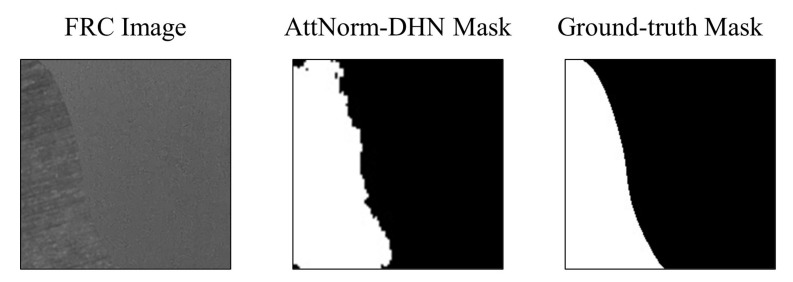
Illustration on the mask predicted by the AttNorm-DHN, along with its corresponding FRC image and ground-truth mask.

**Table 1 micromachines-14-00879-t001:** Network architecture of the encoder network.

Operation	Activation Function	Output Shape
Input image		1 × 200 × 200
Conv 3 × 3	ReLU	64 × 200 × 200
Conv 3 × 3	ReLU	64 × 200 × 200
Maxpool 2 × 2		64 × 100 × 100
Conv 3 × 3	ReLU	128 × 100 × 100
Conv 3 × 3	ReLU	128 × 100 × 100
Conv 1 × 1	ReLU	64 × 100 × 100

**Table 2 micromachines-14-00879-t002:** Network architecture of the attention network.

Operation	Activation Function	Output Shape
Input image		1 × 200 × 200
Conv 3 × 3	ReLU	64 × 200 × 200
Conv 3 × 3	ReLU	64 × 200 × 200
Maxpool 2 × 2		64 × 100 × 100
Conv 3 × 3	ReLU	128 × 100 × 100
Conv 3 × 3	ReLU	128 × 100 × 100
Maxpool 2 × 2		128 × 50 × 50
Conv 3 × 3	ReLU	256 × 50 × 50
Conv 3 × 3	ReLU	256 × 50 × 50
Maxpool 2 × 2		256 × 25 × 25
Conv 3 × 3	ReLU	512 × 25 × 25
Conv 3 × 3	ReLU	512 × 25 × 25
Upsample 2 × 2		512 × 50 × 50
Conv 3 × 3	ReLU	256 × 50 × 50
Conv 3 × 3	ReLU	256 × 50 × 50
Upsample 2 × 2		256 × 100 × 100
Conv 3 × 3	ReLU	128 × 100 × 100
Conv 3 × 3	ReLU	128 × 100 × 100
Conv 1 × 1	ReLU	1 × 100 × 100

**Table 3 micromachines-14-00879-t003:** Network architecture of the Norm-DHT.

Operation	Activation Function	Output Shape
Input features		64 × 100 × 100
Norm-DHT		64 × 101 × 180
Conv 3 × 3	ReLU	64 × 101 × 180
Conv 3 × 3	ReLU	64 × 101 × 180
Conv 1 × 1	ReLU	1 × 101 × 180

**Table 4 micromachines-14-00879-t004:** The evaluation metrics of different methods on three separate datasets.

Method	Normal Dataset			Longline Dataset			Background Dataset		
	F-Measure	MAE	RMSE	F-Measure	MAE	RMSE	F-Measure	MAE	RMSE
Canny+HT [31]	0.705	1.997	9.711	0.0158	5.993	6.883	0.193	26.651	39.458
DHN [11]	0.981	**0.018**	**0.137**	0.97	0.689	6.49	0.714	4.122	14.568
Norm-DHN	0.986	0.022	0.676	**0.984**	**0.015**	**0.126**	0.767	2.191	10.502
AttNorm-DHN	**0.990**	0.07	2.102	0.981	0.241	3.167	**0.855**	**0.239**	**2.401**

**Table 5 micromachines-14-00879-t005:** The evaluation metrics of DHN, Att-DHN, and AttNorm-DHN on the background dataset.

Method	F-Measure	MAE	RMSE
DHN [11]	0.714	4.122	14.568
Att-DHN	0.824	0.468	4.23
AttNorm-DHN	**0.855**	**0.239**	**2.401**

**Table 6 micromachines-14-00879-t006:** The semantic segmentation evaluation metrics of U-net and AttNorm-DHN on the background dataset.

Method	IOU	PA
U-Net [37]	**0.994**	**0.986**
AttNorm-DHN	0.941	0.863

## Data Availability

The data presented in this study are available on request from the corresponding author. The data are not publicly available due to privacy.

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
