# Peer review of "Automated Industrial Composite Fiber Orientation Inspection Using Attention-Based Normalized Deep Hough Network"

_micromachines, 2023, doi:10.3390/mi14040879_

Round 1
Reviewer 1 Report
This paper presents a novel approach for automated fiber orientation inspection in composite manufacturing processes using an attention-based normalized deep Hough network. The main contributions can be summarized as follows.
1) Hough normalization (HN) - a technique that normalizes the accumulated votes in the Hough space by the length of the corresponding line segment. It ensures each line segment contributes equally to the detection process.
2) Attention-based deep Hough network - a framework that integrates an attention network and the Hough network. It is able to selectively focus on important regions of the image, thereby mitigating the impact of background noise caused by mold in FRC images.
3) Establishment of three separate datasets - normal FRC images, FRC images with longline segments noise, and FRC images with background noise. Extensive evaluation of the proposed method achieves competitive performance against state-of-the-art methods in terms of F1, MAE, and RMSE.
This work showcases the frontiers of intelligent precision machining through the integration of artificial intelligence and manufacturing processes. The paper is well-structured, clearly written, and provides a thorough introduction to the problem of fiber orientation inspection in FRC, as well as a detailed description of the proposed method. The experiments are well-designed and the results are convincing. But there are still several concerned issues that should be addressed before publication. These issues are listed as follows.
1) The authors should provide a more detailed explanation of the reasons for the occurrence of these noises, whether they frequently appear in the manufacturing scenario, and why these noises become serious issues in the manufacturing scenario.
2) The paper could provide more details on the datasets used for evaluation, such as the types of FRC materials and molds.
3) It would be helpful if the paper provides more information on the computational requirements of the proposed method.
Reviewer 2 Report
The reviewed paper concerns the analysis of fiber prepreg sheets images in the production of fiber-reinforced composites. Problem appears to be interesting.
The authors propose the use of a Hough transform supplemented with a normalization and a Attention CNN.
In the introduction and abstract, the authors claim to present a novel Hough normalization and an innovative design of attention based deep Hough network.
After short searching in Google scholar service I have found that the normalized Hough transform has been introduced earlier please see for example: KRIEGER LASSEN, N. C. Automatic localisation of electron backscattering pattern bands from Hough transform. Materials Science and Technology, 1996, 12.10: 837-843. Many more are available.
Also attention networks has been parried with Hough transform before please see for example: Wang, Qi, et al. "Multitask attention network for lane detection and fitting." IEEE transactions on neural networks and learning systems 33.3 (2020): 1066-1078.
I do not claim that the presented framework does not contain any novelties, but the two above are questionable.
Relatively fairly is written chapter 2 "related works", but references to normalized Hough transform and attention networks used with Hough transform should be added.
The biggest problem of the presented manuscript are the deficiencies in the description of the materials and methods used.
A very limited description of the research material is provided in section 5.1.1. There is no information about the image acquisition system, such as:
- used cameras,
- resolution of acquired images,
- number of color channels,
- number of bits per pixel,
- type of compression or lack of it,
- ...
There is information about the acquisition of images for 80 types of illuminations without any details provided.
Also only information about image preprocessing is provided in line 329 where resize to the 200x200 pixels are reported. But there is no information here whether the rescaling took place before or after the distortion was applied.
In my opinion, extended information from chapter 5.1.1 should be moved to chapter 4, which should be titled "materials and methods"
Similarly, section 3 should be moved to a subsection in section 4 before description of the methods.
The description of the methods should be created from the scratch. Such a description could start with the presentation of the general concept of the framework shown in Fig. 5. Then detailed descriptions of the individual components of this framework should be provided. At the same time, it is necessary to explain the need for using each of this components. There are 3 convolutional networks in this framework:
- Encoder network
- Attention network
- Normalized Deep Hough Transform network
The following information is missing from the description of these networks:
- the exact structure of each network, including: what are the individual layers in the network, what are the sizes of convolutional filters, what is the number of these filters, type of activation functions, the presence and type of pooling layers, the presence of dropout layers,
- how the weights of the network were obtained, were they obtained from outside, were they trained in-house through transfer learning, or were they learned from scratch.
In section 5.1.3 (lines 327-328) there is an info that: ” The first three layers of VGGNet16 are used as the encoder network” but no details about network structure and how the weights were obtained are provided. In the same section lines 328 and 329 an info can be found that :” The size of the convolutional kernel is 3×3” without info if this refers to the encoder network or to all networks used in the framework.
Some detailed comments:
1) Line 40, reference 3 deals with nondestructive tests while it should support destructive test, please cite something describing destructive tests. Reference [3] better fits next sentence.
2) Line 48, some seminal paper on the Hough transformation should be cited here
3) Although Houg's deep transform is cited on line 181, it would be nice to cite it back in section 4.1.1, otherwise the reader who doesn't read the entire document might get the impression that the deep transform is introduced in this article.
4) There is no reference to the normalized Hough transform in section 4.1.2
5) The paper does not explain the need for an encoder network. One sentence in line 233 and section 4.2.1 does not explain the function of this network.
6) The sentence in line “The texture features of FRC image are distributed on many parallel lines” sounds awkward. It will be better to write it “the texture consists of long parallel lines” Similarly, part of the sentence in lines 188-189 and in the caption of Figure 3 “A straight line l is one representative fiber texture,” should be reformulated. I have the impression that the authors do not fully understand the concept of image texture.
7) The network description from the line 251:” “The attention network firstly adopts the same network structure as the encoding” is inconsistent with the schematic presented in figure 5. Encoder network seems to consists of one convolutional layer followed by pooling layer then followed by two convolutional layer while in the Attention network before bottleneck there are three convolutional layers each followed by pooling layer.
8) Equations 11 and 12 do not account for the 180-degree periodicity of orientation measurements.
9) Table 2 is the part of the table 1 and does not provide any new data.
10) The DOI numbers of most reference articles are missing, for example [4] https://doi.org/10.1007/s12221-016-6049-z ,
Reviewer 3 Report
"Journal Not Specified" at bibliographic detail (and consequential headers)!
The title includes "in Intelligent Industrial Manufacturing Process" but this aspect is not a significant aspect of the paper after superficial treatment in the Introduction!
Abstract first line: "superior needs a reference case?
Abstract final line: "F1, MAE and RMSE" are undefined abbreviations (and likely to be unfamilar to composites personnel)
Lines 22-23: "composites" (plural) should be followed by "their" (not "its")!
Line 24: also excellent corrosion resistance?
Line 26 new sentence: "Resin/polymer impregnated fibre sheets (prepregs)"?
Line 35: "an efficient fiber orientation inspection system .. is desperately needed [if the fiber placement system is not under good control]"
Line 42: thermography would work with other heat sources (not just laser)?
Line 48: the composites manufacturing readership would probably need more information on HT/DHT?
Figure 2 needs labels for the reference line and the gaps.
No discussion of specimen preparation or system lighting?
Are the considered materials simply aligned unidirectional fibres with no waviness or crimp?
Is the system equally applicable to (a) transparent (glass) and opaque (aramid/carbon) fibres, and to the presence of (b) voids and (c) cross-overs in woven fabrics?
Line 226: "all-ones"! meaning a matrix with every pixel set at unity?
line 275: "Both the longline dataset and the background dataset are synthetic datasets" so the processed and analysed images bear little relation to real composites manufacturing systems!
Line 293: delete second "r" from "server".
Line 316: what about true negative (TN)?
Line 376: but the noise is synthetic not real!
Round 2
Reviewer 2 Report
Most of my comments have been incorporated into the revised version of the manuscript.
A few issues that should be addressed are:
1. Line 1: I will exchange “superior” with “excellent” or “superb”.
2. I don't like the term "noise" in phrases: “background noise” and “longline segments noise”. Noise has a pretty strict definition in images that doesn't fit here. Please consider other terms such as "foreign elements in the image", "damage to the structure of prepreg sheets".
3. I still strongly suggest removing the word "novel" from line 9 and from the rest of the text. The proposed method is identical to the DHT normalization method is identical to the popular HT normalization method. The absence of the word "novel" does not diminish the importance of the scientific work presented here.
4. Similarly, I would remove the word "innovatively" from line 12.
5. Add maxpool and upsample sizes in tables 1-3.
6. In equations 11 and 12, a simpler error expression can be used:
e = acos(abs(cos(alfa – beta)))
This expression is differentiable unlike the one used in equations 11 and 12
7. After the introduced corrections, the text still lacks clear information which elements of the framework were trained during the work. I suppose all of them. But it is nowhere explicitly stated.
8. Even though I'm not a native speaker, I strongly suggest using a professional proof reading service. The style in which the manuscript is written can be significantly improved. Especially since the substantive work is now written in an acceptable way. (Many sentences are too long. Better wording could be used in many places. See points 1 and 2)
Reviewer 3 Report
Revised manuscript almost good to publish, but would benefit from:
Lines 197 and 204: remove initials before author names.
Line 267: mismatched brackets/parentheses?
Line 299: reference for ReLu?
Line 300: reference for Maxpooling?
I do not wish to see the revision.
